# DFT Study on the CO_2_ Reduction to C_2_ Chemicals Catalyzed by Fe and Co Clusters Supported on N-Doped Carbon

**DOI:** 10.3390/nano12132239

**Published:** 2022-06-29

**Authors:** Qian Xue, Xueqiang Qi, Tingting Yang, Jinxia Jiang, Qi Zhou, Chuang Fu, Na Yang

**Affiliations:** 1School of Chemistry and Chemical Engineering, Chongqing University of Technology, Chongqing 400054, China; xueq@stu.cqut.edu.cn (Q.X.); Tingty@stu.cqut.edu.cn (T.Y.); QiZhou@stu.cqut.edu.cn (Q.Z.); ChuangFu@stu.cqut.edu.cn (C.F.); 2Chongqing Key Laboratory of Chemical Process for Clean Energy and Resource Utilization, School of Chemistry and Chemical Engineering, Chongqing University, Chongqing 400044, China; 3Chongqing Medical and Pharmaceutical College, Chongqing 400020, China

**Keywords:** DFT, CO_2_RR, clusters, selectivity and activity

## Abstract

The catalytic conversion of CO_2_ to C_2_ products through the CO_2_ reduction reaction (CO_2_RR) offers the possibility of preparing carbon-based fuels and valuable chemicals in a sustainable way. Herein, various Fe_n_ and Co_5_ clusters are designed to screen out the good catalysts with reasonable stability, as well as high activity and selectivity for either C_2_H_4_ or CH_3_CH_2_OH generation through density functional theory (DFT) calculations. The binding energy and cohesive energy calculations show that both Fe_5_ and Co_5_ clusters can adsorb stably on the N-doped carbon (NC) with one metal atom anchored at the center of the defected hole via a classical MN_4_ structure. The proposed reaction pathway demonstrates that the Fe_5_-NC cluster has better activity than Co_5_-NC, since the carbon–carbon coupling reaction is the potential determining step (PDS), and the free energy change is 0.22 eV lower in the Fe_5_-NC cluster than that in Co_5_-NC. However, Co_5_-NC shows a better selectivity towards C_2_H_4_ since the hydrogenation of CH_2_CHO to CH_3_CHO becomes the PDS, and the free energy change is 1.08 eV, which is 0.07 eV higher than that in the C-C coupling step. The larger discrepancy of d band center and density of states (DOS) between the topmost Fe and sub-layer Fe may account for the lower free energy change in the C-C coupling reaction. Our theoretical insights propose an explicit indication for designing new catalysts based on the transition metal (TM) clusters supported on N-doped carbon for multi-hydrocarbon synthesis through systematically analyzing the stability of the metal clusters, the electronic structure of the critical intermediates and the energy profiles during the CO_2_RR.

## 1. Introduction

The electrochemical CO_2_ reduction reaction (CO_2_RR), as a useful method to convert CO_2_ into value-added chemical products, which not only helps to solve the energy and environmental problems caused by fossil fuel combustion but also achieves sustainable development [1,2,3,4]. The main products of CO_2_RR are generally divided into C_1_ products (e.g., CO, CH_4_, CH_3_OH, HCOOH, etc.) [5] and C_2_ products (e.g., C_2_H_4_, C_2_H_5_OH, CH_3_COOH, etc.) [6]. Cu and Cu-derived materials have been considered the most common electrocatalysts for the CO_2_RR in the early stages [7,8]. Furthermore, Ag-based [9,10] and Au-based [11,12,13] catalysts can selectively reduce CO_2_ to CO at low overpotentials. However, they suffer from low utilization of metal atoms and a low C_2+_ selectivity. 

Recently, the single-atom catalysts (SACs) of metal loaded on carbon substrates (metal nitrogen-doped carbon-based catalysts) have become a rather hot frontier for the maximized atom utilization efficiency and defined active centers. Rossmeisl et al. [14,15] found that the transition metal nitrogen-doped carbon-based catalysts (M-N-C, M = Mn, Fe, Co, Ni or Cu) performed a high CO selectivity for CO_2_RR. Furthermore, both Mn-N-C and Fe-N-C also possessed CO selectivity as well as trace amounts of CH_4_, which was assigned to the stronger CO binding of the Fe and Mn porphyrine-like structures. Zu et al. [16] successfully synthesized atomically dispersed Sn sites on nitrogen-doped carbon, which performs excellent activity and stability for formate generation at a kilogram scale with a quick freeze-vacuum drying-calcination method. Many Ni-based, Fe-based and Co-based SACs have exhibited high electrocatalytic activity and Faradaic effectivity (FE) for the CO_2_RR with CO as the primary product due to the moderate adsorption energies of *COOH and *CO intermediates, as well as the high activation barrier for the hydrogen evolution reaction (HER) [17,18].

Though the widespread study on the single-atom catalysts enhanced the utilization efficiency of metal atoms, most of the current studies are limited to the reduction of CO_2_ to C_1_ products. Compared to C_1_ products, C_2+_ products have a higher economic and chemical utilization value [19,20,21]. Cu-based SACs, up to now, have still performed good electrochemical reduction of CO_2_ to C_2+_ chemicals [22,23]. However, Karapinar [24] revealed that the atomically dispersed CuN_x_ sites could reversibly convert into Cu clusters during CO_2_RR, which are suggested as the real multiple active sites for CH_3_CH_2_OH production. Considering the fact that a single metal atom can accommodate only a single CO, it is difficult to activate two CO_2_ molecules simultaneously to trigger the C-C coupling reaction based on an isolated metal center. Thus C-C coupling proceeding on the single metal atoms is quite difficult. Therefore, the catalysts with multiple active sites need to be considered to achieve the conversion from CO_2_ to C_2_ products [25].

Transition metal (TM) clusters with precise atomic numbers can offer multiple active sites, tune the size-dependent catalytic activity [26], and allow them to find the highest reactivity for the activation and dissociation of strong chemical bonds from CO_2_. Xu et al. [27] reported a facile underpotential deposition technique to fabricate Cu clusters on carbonaceous substrates via rationally introducing S dopants in graphite foam. The obtained free-standing electrode exhibited high activity and excellent long-term stability toward oxygen reduction reaction. Pei et al. [28] found the trimeric metal clusters anchored on N-doped porous graphitic sheets possess a good selectivity and superiority towards CO_2_RR to multi-carbon products due to the multiple active sites.

Considering the loading of metal clusters with a precise number of atoms on the active substrate can not only avoid the problem of low stability of bare metal cluster catalysts at room temperature but also further enhance their stability and catalytic efficiency. Graphite-based materials are currently widely used as substrates for electrocatalysts. To access C_2_ products more efficiently, herein, we employed density functional theory (DFT) calculations to explore the CO_2_RR catalyzed by Fe_n_ (*n* = 1, 3–5) anchoring on N-doped carbon (Fe_n_-NC) to C_2_H_4_ and C_2_H_5_OH in this work. Furthermore, the Co_5_ cluster supported on N-doped carbon (Co_5_-NC) was explored comparatively. We found that Fe_5_ loaded on NC exhibit significant activity for promoting the reduction of CO_2_ to C_2_ products, while the Co_5_ cluster has higher priority for the selective synthesis of C_2_H_4_. Our findings provide insights into the design of highly active catalysts for CO_2_RR and create a platform for developing metal cluster-NC electrocatalysts. 

## 2. Theoretical Method

First principle calculations were performed using DFT with spin polarization utilizing the Vienna Ab initio Simulation Package (VASP). The projected augmented wave (PAW) [29,30,31] was used, and the generalized gradient approximation (GGA) realized by the Perdew–Burke–Ernzerhof function (PBE) was adopted to incorporate the exchange-correction functional [32]. A 2 × 2 × 1 Monkhorst-Pack K-point was sampled in the Brillouin zone, and a cut-off energy of 500 eV was set for geometric optimization. The convergence criteria are of 10^−5^ eV in energy between two electronic iteration steps and 0.02 eV/Å in force for every atom [33]. Our calculations of catalytic performance are based on the computational hydrogen electrode (CHE) proposed by Nørskov et al. [34]:(1)$$H_{\left(aq\right)}^{+}+e^{-}\to \frac{1}{2}H_{2}\left(g\right)$$

The change of the free energy for the step $*A+H^{+}+e^{-}\to *AH$ can be equal to the reaction: $*A+\frac{1}{2}H_{2}\left(g\right)\to *AH$ at 0 V versus the reversible hydrogen electrode (RHE) at all pH values.

We employ five types of small iron clusters Fe_n_ (*n* = 1, 3–5) supported on nitrogen-doped carbon sheets as the calculation models. The defects of *NC* provide ideal anchor sites for the iron cluster. To estimate the stability of supported TM clusters, the binding energy (*E_b_*) of *TM_n_* cluster on *NC* is calculated by Equation (2)
(2)$$E_{b}=E_{TM_{n}-NC}-E_{TM_{n}}-E_{NC}$$

Here $E_{TM_{n}-NC}$ is the total energy of the optimized *TM_n_* cluster supported on *NC*. The terms $E_{TM_{n}}$ and $E_{NC}$ refer to the energies of isolated *TM_n_* cluster and support. We calculated the cohesive energy (*E_c_*) of each *TM_n_* cluster to further evaluate the stability of *TM_n_-NC* catalysts, with *E_c_* defined as:(3)$$E_{c}=\left(E_{cluster}-nE_{TM}\right)/n$$

Here the $E_{cluster}$ and $E_{TM}$ represent the energy of the total energy of the *TM_n_* cluster and the energy of single *TM* atom; n is the number of TM atoms in the cluster. The more negative cohesive energy (*E_c_*) indicates a more stable structure. 

The adsorption energy (*E_ads_*) of every intermediate species is defined by Equation (4)
(4)$$E_{ads}=E_{C_{x}H_{y}O_{z}-TM_{n}-NC}-E_{TM_{n}-NC}-E_{C_{x}H_{y}O_{z}}$$
where $E_{C_{x}H_{y}O_{z}-TM_{n}-NC}$ refers to the total energy of the adsorbed species on the supported TM_n_ cluster, and $E_{TM_{n}-NC}$ is the energy of supported TM_n_ cluster. $E_{C_{x}H_{y}O_{z}}$ refers to the energies of $C_{x}H_{y}O_{z}$ in gas phase, respectively. The more negative adsorption energy indicates a stronger binding between *TM* cluster and *NC* support. Gibbs free energy change (ΔG) [35,36] is defined as: (5)$$\Delta G=\Delta E+\Delta E_{ZPE}+\Delta \int C_{P}dT-T\Delta S+\Delta G_{pH}+\Delta G_{U}$$
where ΔE, $\Delta E_{ZPE}$, $\Delta \int C_{P}dT$ and ΔS are the total energy difference, the zero-point energy difference, the difference in enthalpic correction and the entropy change between the products and reactants obtained from DFT calculations, respectively. The zero-point energies (*ZPE*) and total entropies of the gas phase were computed from the vibrational frequencies, and the vibrational frequencies of the adsorbed species were also computed to obtain the *ZPE* contribution to the free energy expression. Only vibrational modes of the adsorbates were computed explicitly, while the catalyst sheet was fixed (assuming that vibration contribution to the free energy from the substrate is negligible) [37,38]. T is the temperature (298.15 K). The influence of applied potential is: $\Delta G_{U}=-neU$, where U is the external potential versus RHE, e is the electron transfer, and n is the number of proton–electron pairs. $\Delta G_{pH}$ is the free energy correction due to the concentration of H^+^. $\Delta G_{pH}=-k_{B}T\times \ln\left[H^{+}\right]=k_{B}T\times \ln10\times pH$, where k_B_ is the Boltzmann constant, and the value of pH was assumed to be zero for acidic conditions.

## 3. Results and Discussion

### 3.1. The STABILITY Analysis of Fe_n_-NC

For the Fe_n_ clusters supported on the NC substrate (Fe_n_-NC), *n* = 1, 3, 4 and 5 were chosen to be studied here since the Fe_2_ cluster is unstable [39]. As shown in Figure 1, for the adsorption of a single Fe atom on the NC substrate (Figure 1a), the mono Fe atom coordinated with the four nitrogen atoms and Fe-NC maintains a perfect monolayer structure, which is in agreement with previous results [40]. For the adsorption of Fe_n_ clusters with n ranging from 3 to 5 (Figure 1b–d), one Fe atom is anchored at the same position with that in a single Fe atom. Two Fe atoms bound to the doped nitrogen atoms with distances of about 2.1~2.3 Å, respectively, while the other Fe atoms bound together through Fe-Fe metal bonds. 

As listed in Table 1, all the binding energies of Fe_n_ clusters on NC supports are thermodynamically favorable (E_b_ < 0). With the increase in Fe atoms, the binding energy decreases except for the magic Fe_5_ cluster, which means that the small Fe clusters may tend to aggregate from small clusters to bigger clusters on NC support. The reason for the decreased binding energy of the magic Fe_5_ cluster lies in that the Fe^5site^ located at the top site, as shown in Figure 1d. Hence there is no interaction with the NC support. What is more, the cohesive energy of various Fe_n_ clusters was also explored according to Equation (3), as shown in Table 1. It can be found that with the increase in Fe atoms in the cluster, the cohesive energy becomes thermodynamically favorable.

### 3.2. Electrocatalytic CO_2_RR

The charge difference between two active transition atoms plays a key role during CO_2_RR, and the mixed oxidation state of the catalytic centers can boost the C-C coupling [41,42]. Thus the Bader charges of various Fe clusters adsorbed on NC were investigated. As listed in Table 1, one can find that the electrons can transfer from the clusters to the support, making the whole Fe cluster positively charged, and each Fe atom in the cluster is positively charged as well. Herein, the Fe_n_ clusters are favorable to the CO_2_RR; thus the electron-accepting properties of the positively charged Fe and Co sites are favorable for stabilizing the CO_2_RR intermediates [43]. The largest charge transfer can be determined for the single Fe atom configuration since the monatomic Fe interacts with four coordinated N atoms. Furthermore, significant discrepancies for each charged Fe atom in the Fe_3_, Fe_4_ and Fe_5_ clusters can be determined, which is beneficial for the C-C coupling reaction.

The conversion of CO_2_ to CO catalyzed by various Fe_n_ clusters was calculated, as shown in Figure 2; the PDS is the *CO to CO for all the Fe_n_-NC with a maximum Gibbs free energy ($\Delta G_{3}=\Delta G_{CO}-\Delta G_{*CO}$). The ΔG_3_ are 0.91, 1.14 and 1.58 eV for the Fe-NC, Fe_3_-NC and Fe_4_-NC, respectively. This value increases to 1.94 and 2.29 eV at two different sites of Fe_5_-NC. Thus it can be deemed that the Fe-NC, Fe_3_-NC and Fe_4_-NC require a lower overpotential to drive the desorption of CO, indicating that these structures favor the conversion of CO_2_ to CO. The $*CO\to CO\left(g\right)$ step with strong *CO binding leads to a positive ΔG of CO desorption. The relatively strong binding of *CO on Fe-N_x_ is fully supported by the experimentally confirmed exclusive ability of the Fe-N_x_ catalyst to produce the hydrocarbon CH_4_ [44].

In simple terms, one could say that to produce subsequent reaction products from CO during the CO_2_RR; the CO molecule must be bound strong and long enough to undergo subsequent dissociation and hydrogenation steps to arrive at CH_4_ or other small organic molecules. Herein, our work focuses on the reduction of CO_2_ to C_2_ products, which requires improved selectivity and activity by inhibiting the unwanted, i.e., C_1_ hydrocarbon reaction pathway, which favors both the stabilization of *CO on the catalyst surface and the formation of C-C bonds. The strongest interaction for CO on the two Fe_5_-NC sites means that CO does not leave the iron cluster surface easily, which, in turn, favors the subsequent C_2_ product conversion. Therefore, the following study focuses on the performance of electrocatalytic CO_2_ reduction to C_2_H_4_ and CH_3_CH_2_OH over the Fe_5_-NC cluster. In order to further extend this result to other systems, the Co_5_ cluster supported on NC was chosen to be studied comparatively (Appendix A). As shown in Table 2, the E_b_ value between the Co_5_ cluster and the NC is −10.00 eV, and the E_c_ value of the Co atom is −1.39 eV, which means the Co_5_ cluster can adsorb stably on the NC.

To further study the mechanisms of CO_2_RR catalyzed by Fe_5_-NC and Co_5_-NC, the optimized structures and the energy profiles along the reaction coordinate for CO_2_RR to both C_2_H_4_ and CH_3_CH_2_OH on Fe_5_-NC and Co_5_-NC are calculated as shown in Figure 3 and Figure 4, respectively. It can be found that two strongly adsorbed CO molecules adsorbed on the two adjacent metal atoms through carbon atoms either on the Fe_5_ or the Co_5_ clusters before the C-C bond formation. The two CO molecules will couple with each other via the top Fe or Co atom in the following steps of CO_2_RR. For the CO_2_RR on Fe_5_-NC, the Gibbs free energy for the hydrogenation of the *CO dimer is uphill with an energy value of 0.79 eV, which is the highest energy during the formation of both C_2_H_4_ and CH_3_CH_2_OH. Thus, it can be deemed that the C-C coupling reaction for the CO_2_RR from CO_2_ to C_2_ chemicals is the PDS. Furthermore, it can be speculated that both C_2_H_4_ and CH_3_CH_2_OH products can be achieved with Fe_5_-NC catalyst, and the amount of C_2_H_4_ should be much more than the CH_3_CH_2_OH. Because the Gibbs free energy for the hydrogenation of *CH_2_CHO to C_2_H_4_ is energy thermodynamically favorable, while the hydrogenation of *CH_2_CHO to *CH_3_CHO is an uphill reaction with a free energy of 0.26 eV. 

For the CO_2_RR on Co_5_-NC, the Gibbs free energy for the hydrogenation of the *CO dimer is uphill with an energy of 1.01 eV, which is 0.22 eV higher than that of the Fe_5_-NC. However, the Gibbs free energy for the hydrogenation of *CH_2_CHO to *CH_3_CHO is (0.07 eV higher) comparable with that in the C-C coupling reaction, which means that the hydrogenation of *CH_2_CHO to *CH_3_CHO becomes the PDS. Thus, most of the final C_2_ products should be C_2_H_4_. In general, the Fe_5_-NC has good catalytic activity towards the C_2_ chemicals with relatively lower free energy change (0.53 eV), while the selectivity is not as good as Co_5_-NC. However, Co_5_-NC possesses better selectivity while the activity is lower than Fe_5_-NC.

The d-band center of the TM and its electronic occupancy can affect the bonding strength between the intermediate and the catalytic surface. As shown in Figure 5, the PDOS of Fe *d* orbit from the top- and sub-layer structures show much more differences than that of the Co *d* orbit on Co_5_-NC. Furthermore, the d band center of the top Fe atom in the Fe_5_ cluster is −3.48 eV, while it becomes −1.59 eV for the sub-layer atoms. However, the d band center of the top Co atom is −1.34 eV, and it only changes to −1.47 eV for the sub-layer atoms. A much bigger discrepancy of the d band center between the top Fe atom and sub-layer atoms than that in the Co_5_ cluster may boost the C-C coupling reaction, which could be called the synergy effect between the top- and sub-layer metal atoms. Our findings are consistent with the synergy effect between Cu^+^ and Cu^0^, and the surface can significantly improve the kinetics and thermodynamics of both CO_2_ activation and CO dimerization. Cu metal embedded in an oxidized matrix catalyst can promote CO_2_ activation and CO dimerization for electrochemical reduction of CO_2_ [41].

Our theoretical calculations found that the multiple active sites in both the Fe_5_ and Co_5_ cluster-based catalysts facilitate the stabilization of *CO on the catalyst surface and the formation of C-C bonds. Both geometrical effects and electronic effects are the key factors leading to the Fe_5_ and Co_5_ clusters exhibiting better activity and/or selectivity over the single metal component. Furthermore, the tunable synthesis of Fe and Co alloys supported on NC may promote both their activity and selectivity toward CO_2_RR. Therefore, Fe_5_, Co_5_ and the related tunable alloy clusters show great potential applications in electrocatalytic CO_2_RR, and our methods provide a concept for designing the improved CO_2_RR electrocatalysts.

## 4. Conclusions

The stability of the Fe_n_ (*n* = 1, 3, 4, and 5) clusters was studied first, and it can be found that the Fe_n_ anchors stably on the nitrogen-doped carbon via a basic Fe-NC structure. With the increasing of Fe atoms in the cluster, both the binding energy and cohesive energy become thermodynamically favorable, which means a small cluster tends to aggregate to be a bigger one. While for the Fe_5_ cluster, the binding energy decreases because there is no interaction between the topmost Fe atom with the NC support anymore. In addition, the CO desorption is the most difficult on the Fe_5_ cluster, which is beneficial to the subsequent reaction products from CO. Hence, the Fe_5_-NC cluster was chosen to be studied as our C_2_ catalyst, and Co_5_-NC was comparatively studied as well. The results show that Fe_5_-NC has better activity towards CO_2_RR, and the products should be the mixed C_2_H_4_ and CH_3_CH_2_OH, since the PDS is the C-C coupling reaction with a free energy change of only 0.79 eV. The free energy change is only 0.53 eV for the reduction of CH_2_CHO to CH_3_CHO, and the reduction of CH_2_CHO to C_2_H_4_ is a spontaneous step without any free energy change. Considering the fact that C_2_H_4_ is a gas, Fe_5_-NC should be a good catalyst for CO_2_RR to liquid ethanol with a relatively lower yield since part of the C_2_H_4_ gas will also be produced. Furthermore, Co_5_-NC possesses a relatively good selectivity, but bad activity since the reduction of CH_2_CHO to CH_3_CHO is the PDS, and the free energy change is 1.09 eV. The PDOS and d band center analysis demonstrates that the relative energy favorable C-C coupling reaction on the Fe_5_ cluster could be attributed to the larger discrepancy of d electrons of the two CO-adsorbing Fe atoms. This paper predicts a good application prospect of TM clusters supported on nitrogen-doped graphene for CO_2_RR, and the new insight into the relationship between selectivity and activity sheds light on a new route for understanding and designing highly efficient non-precious catalysts for CO_2_RR.

## Figures and Tables

**Figure 1 nanomaterials-12-02239-f001:**
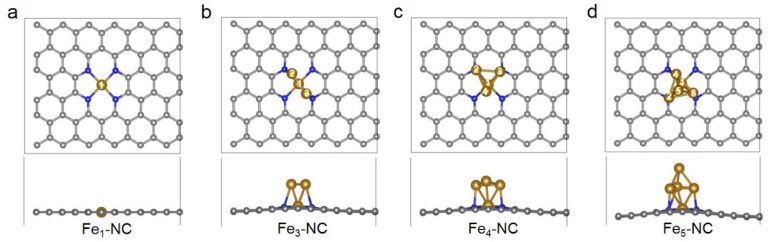
Top-view and side-view of optimized structures of Fe_1_ (**a**), Fe_3_ (**b**), Fe_4_ (**c**) and Fe_5_ (**d**) clusters supported on four nitrogen-doped carbon. The gold, blue and gray spheres represent Fe, N and C atoms, respectively. The different Fe sites are marked with white numbers.

**Figure 2 nanomaterials-12-02239-f002:**
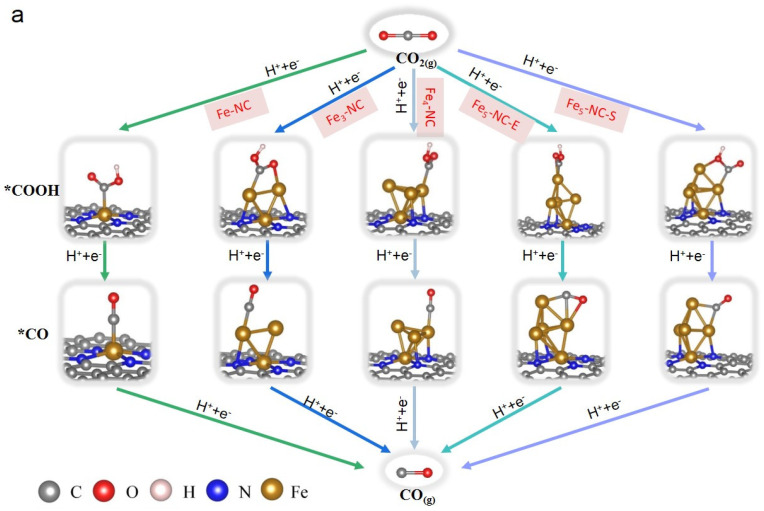
(**a**) The optimized structures of each intermediate during the CO_2_ reduction reaction from CO_2_ to CO, and (**b**) the Gibbs free energy profiles of CO_2_ reduction to CO on Fe_n_-NC catalysts during CO_2_RR.

**Figure 3 nanomaterials-12-02239-f003:**
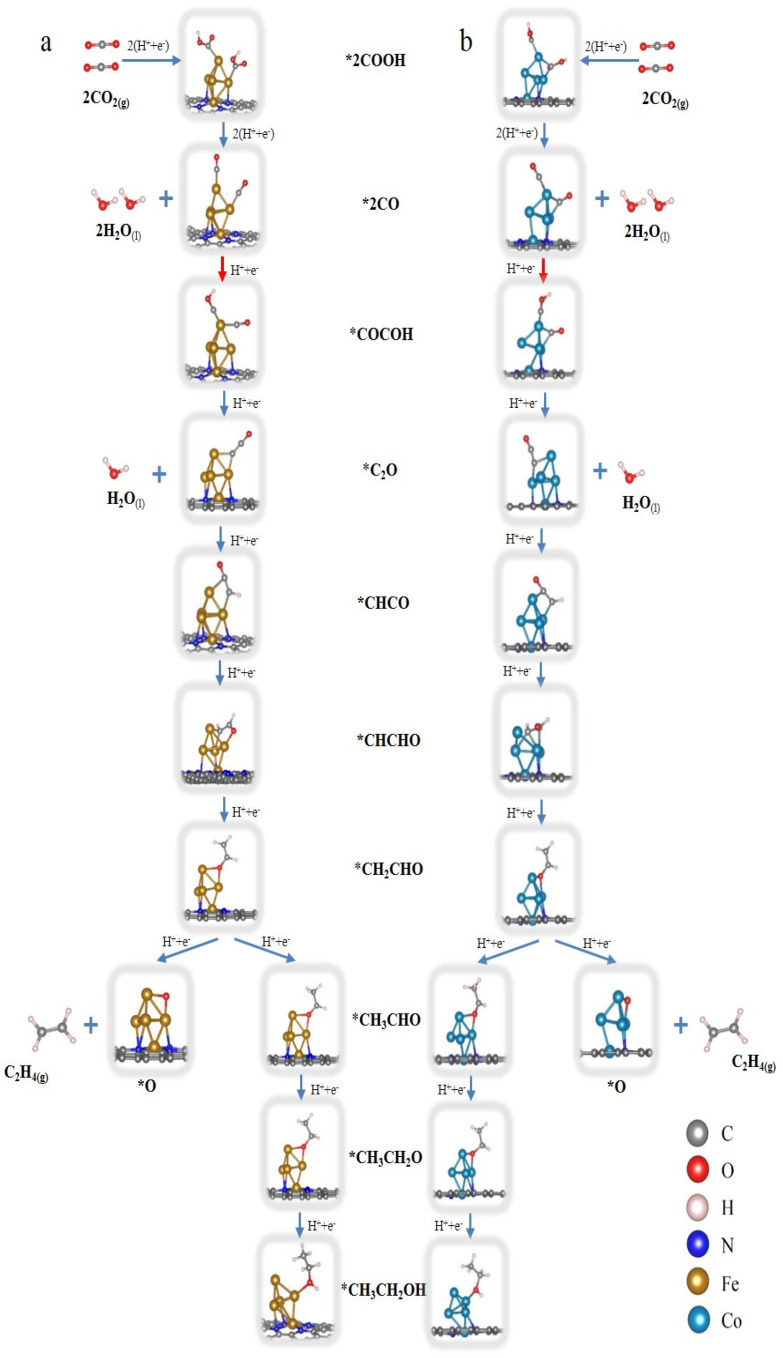
Optimized structures of intermediate Fe_5_-NC (**a**) and Co_5_-NC catalysts (**b**) during the CO_2_RR process.

**Figure 4 nanomaterials-12-02239-f004:**
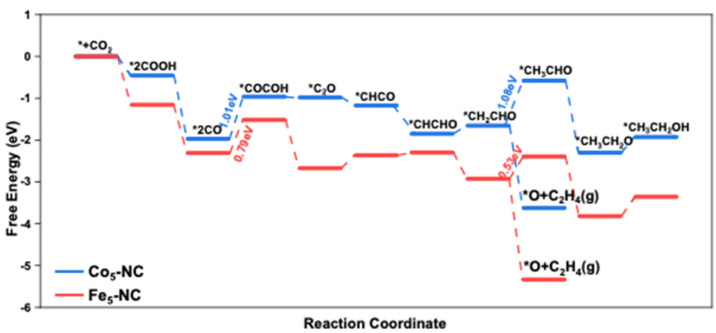
Gibbs free energy profiles for CO_2_RR on Fe_5_-NC and Co_5_-NC.

**Figure 5 nanomaterials-12-02239-f005:**
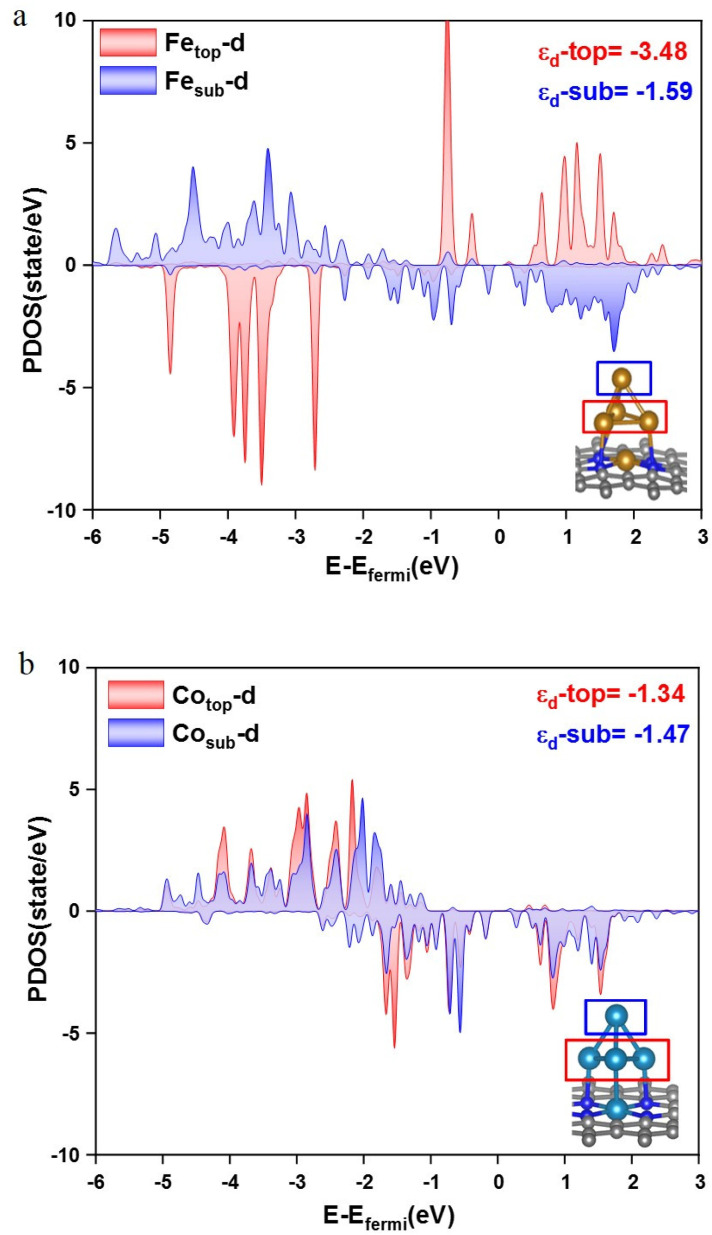
The PDOS of the d orbitals of the single top Fe atom (red) and average three middle Fe atoms (blue) in Fe_5_-NC (**a**), and the PDOS of the d orbitals of the single top Co atom (red) and average three middle Co atoms (blue) in Co_5_-NC (**b**).

**Table 1 nanomaterials-12-02239-t001:** The Bader charge of Fe atom in various Fe_n_-NC, binding energies (*E_b_*) between the Fe_n_ cluster and NC, and the cohesive energy (*E_c_*) of Fe atoms in various Fe_n_ clusters.

Catalyst	* Bader Charge (e)					E_b_ (eV)	E_c_ (eV)
	Fe ^1site^	Fe ^2site^	Fe ^3site^	Fe ^4site^	Fe ^5site^		
Fe_1_-NC	−1.07	-	-	-	-	−9.01	0
Fe_3_-NC	−0.80	−0.32	−0.31	-	-	−9.73	−0.35
Fe_4_-NC	−0.81	−0.34	−0.33	−0.19	-	−9.79	−0.62
Fe_5_-NC	−0.81	−0.36	−0.35	−0.28	−0.01	−8.34	−1.01

* negative Bader charge means electron loss.

**Table 2 nanomaterials-12-02239-t002:** The Bader charge of Co atoms in Co_5_-NC, binding energy (*E_b_*) between the Co_5_ cluster and NC and the cohesive energy (*E_c_*) of Co atoms in the Co_5_ cluster.

Catalyst	Bader Charge (e)					*E_b_* (eV)	*E_c_* (eV)
	Co ^1site^	Co ^2site^	Co ^3site^	Co ^4site^	Co ^5site^		
Co_5_-NC	−0.71	−0.34	−0.31	−0.24	0.06	−10.00	−1.39

## Data Availability

The data presented in this study are available in this article.

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
