# Peer review of "DFT Study on the CO2 Reduction to C2 Chemicals Catalyzed by Fe and Co Clusters Supported on N-Doped Carbon"

_nanomaterials, 2022, doi:10.3390/nano12132239_

Round 1

Reviewer 1 Report

The authors reported the catalytic conversion of CO2 to C2 products via Fen and Co5 catalysts  through density functional theory (DFT) calculations. 

The binding energy (Eb) between Fen cluster and NC, and cohesive energy (Ec) of Fe atoms in various Fen clusters of Fe4-NC, Fe5-NC and, Co5-NC are -9.79 eV vs -0.62 eV, -8.34 eV vs -1.01 eV, and -10,00 eV vs -1.39 eV, respectively. So, Fe in Fe4-NC, Fe5-NC and Co in Co5-NC should be preferred to have bonds with N than with other Fe or Co atom. So, can the authors explain why they choose the model of Fe4-NC, Fe5-NC and Co5-NC in which 1 atom of Fe and Co only have bonds with other Fe or Co without the present of bond with N. 

The author didn't mention about the synthesis study of Fen-NC and Co5-NC. They should review the synthesis method of the layer control of Fe and Co on the support.

Can the author do experiments to compare the results with the calculation data?

Author Response

Thank you so much for your comments. And the responses have been listed in  the attachment.

Reviewer 2 Report

The presented paper is aimed on the theoretical study for designing new catalysts based on the transition metal (TM) clusters supported on N-doped carbon in multi-hydrocarbon synthesis through density functional theory (DFT) calculations. The authors analyzed the stability of metal clusters Fe5 and Co5, their electronic structure and of the critical intermediates and the energy profiles during the CO2RR reaction pathway were made. It was shown that both TM-clusters can be stabilized on the N-doped carbon (with one metal atom anchored at the center of defected hole via a classical MN4 structure. The results obtained demonstrated that Fe5-NC clusters performed better activity than Co5-NC ones, since the carbon-carbon coupling reaction was the potential determining step and the free energy change is 0.22 eV lower on Fe5-NC cluster than that on Co5-NC. However, Co5-NC clusters showed a better selectivity towards C2H4 and the free energy change is 1.08 eV which is 0.07 eV higher than that in the C-C coupling step.

The results obtained in the paper are important for screening and design of  the improved catalysts with high stability, high activity and selectivity for either C2H4 or CH3CH2OH production. The paper can be published in present form after some English improvements and corrections.

Author Response

Thank you so much for your comments. We do some modifications in the revised manuscript.

Reviewer 3 Report

The paper titled: "DFT Study on the CO2 reduction to C2 chemicals catalyzed by Fe and Co clusters supported on N-doped carbon” reports an interesting DFT study finalized to rationalize the differences in activity and selectivity towards CO2 reduction to C2 species catalyzed by Fe and Co clusters supported on N-doped carbon.

The last part of the study seems particularly interesting as correlates the different catalytic properties of the two catalysts to the PDOS of their d orbitals.

However the Authors should pay attention to the sentence at pag. 6: “One can find that the electrons can transfer from the clusters to the support, making the whole Fe clusters positively charged and each Fe atom in the cluster is positively charged as well” while in Table 1 and Table 2 the Bader charge is reported as negative. Do the negative charges refer to the charge transferred to the substrate ?

Author Response

Thank you for your comments. You are right, the negative Bader charge means the electron loss, so the negative charges refer to the charge transferred to the substrate. And the related illustration has been depicted in our manuscript, and you can find the * part (the negative Bader charge means the electron loss) behind the Table 1. We wish it will not take confusion to the readers. 

Round 2

Reviewer 1 Report

I think the quality of the manuscript is good enough for publication.